

# Characterization of Gas and Particle Emissions from Open Burning of Household Solid Waste

Xiaoliang Wang[1], Hatef Firouzkouhi[1], Judith C. Chow[1], John G. Watson[1], Warren Carter[2], Alexandra S.M. De Vos[2]

[1] Division of Atmospheric Sciences, Desert Research Institute, Reno, NV 89512, U.S.A.
       [2] SASOL Research and Technology, Sasolburg, South Africa

*Correspondence to*: Xiaoliang Wang (xiaoliang.wang@dri.edu)

**Abstract.** Open burning of household and municipal solid waste is a common practice in many developing countries. Due to limited resources for collection and proper disposal, solid waste is often disposed of in neighborhoods and open burned in piles

to reduce odors and create space for incoming waste. Emissions from these ground-level and low-temperature burns cause air pollution, leading to adverse health effects among community residents. This study conducted laboratory combustion experiments to characterize gas and particle emissions from ten waste categories representative of those burned in South Africa: paper, leather/rubber, textiles, plastic bottles, plastic bags, vegetation (with three different moisture content levels), food discards, and combined materials. Carbon dioxide ($CO_2$) and carbon monoxide (CO) were measured in real-time to calculate

modified combustion efficiencies (MCE). MCE is used along with video observations to determine fuel-based emission factors (EFs) during flaming and smoldering phases as well as the entire combustion process. Fuel elemental composition and moisture content have strong influences on emissions. Plastic bags have the highest carbon content and the highest combustion efficiency, leading to the highest EFs for $CO_2$. Textiles have the highest nitrogen and sulfur contents, resulting in the highest EFs for nitrogen oxides ($NO_x$) and sulfur dioxide ($SO_2$). Emissions are similar for vegetation with 0% and 20% moisture

contents; however, EFs for CO and particulate matter (PM) from the vegetation with 50% moisture content are 3 and 30 times, respectively, emissions from 0% and 20% moisture contents. This study also shows that neglecting carbon in the ash and PM can lead to significant overestimation of EFs. Results from this study are applicable to emission inventory improvements as well as air quality management to assess the health and climate effects of household waste open burning.



## 1 Introduction

Solid waste management is a global environmental challenge. Approximately two billion metric tons per year of household and municipal solid waste (MSW) are generated globally (Wilson and Velis, 2015). Even though high-income countries have higher per capita MSW generation, waste generation in middle- and low-income countries is growing rapidly due to population growth and economic development (Ferronato and Torretta, 2019). Waste disposal practices include collection, recycling, land filling, incineration, and open burning (Wilson and Velis, 2015; Wiedinmyer et al., 2014). In contrast to the near 100% collection and controlled disposal rates in high and upper-middle income countries, low-income countries often have less than 50% collection rates, with near 0% controlled disposal common in rural areas. It is estimated that at least two billion people worldwide still lack access to solid waste collection, treatment, or disposal services and infrastructure (Cook and Velis, 2021; Wilson et al., 2015).

In rural communities of developing countries, particularly regions where waste collection service is expensive, unavailable, or infrequent, uncontrolled open burning of household solid waste is a common practice for decreasing MSW mass and volume, reducing unpleasant odors from decomposing materials, fueling heating and cooking activities, and destroying pathogens (Cook and Velis, 2021). Globally, about half of the household waste (i.e., about one billion tons) is burned in open, uncontrolled fires every year. Open burning is conducted not only by community members, but also by municipal authorities.

Although perceived as a cost-effective method of waste disposal, uncontrolled solid waste open burning generates a wide range of hazardous substances that pose threats to human health and contribute to climate change (Wiedinmyer et al., 2014; Lemieux et al., 2004). These air contaminants include criteria pollutants, such as carbon monoxide (CO), nitrogen dioxide ($NO_2$), sulfur dioxide ($SO_2$), particulate matter with aerodynamic diameter $\leq 2.5$ µm ($PM_{2.5}$) and $\leq 10$ µm ($PM_{10}$), and lead. Burning also emits other air toxics, such as heavy metal elements, polycholorinated and polybrominated dioxins and furans, and polycyclic aromatic hydrocarbons (PAHs) (Velis and Cook, 2021; Wiedinmyer et al., 2014). Many of these pollutants are carcinogenic or mutagenic; they may cause immunological and developmental impairments and lead to respirable and cardiovascular diseases. It is estimated that exposure to $PM_{2.5}$ from open burning of solid waste causes at least 270,000 premature deaths in the world every year (Williams et al., 2019; Kodros et al., 2016). In addition, open burning emits large amounts of carbon dioxide ($CO_2$) and light absorbing carbon (including black carbon [BC]), two of the largest climate forcers to global warming (Bond et al., 2013; Ipcc, 2013).

Several factors exacerbate the risks of open burning smoke. First, solid waste often contains fuels that release toxic compounds upon heating. For example, construction timber combustion can release high concentrations of arsenic, chromium, and copper. Plastic burning can release phthalates, PAHs, and dioxins (Velis and Cook, 2021; Wasson et al., 2005). Second, the open waste combustion temperatures are typically lower than those in controlled incinerations, resulting in lower combustion efficiencies that generate more pollutants. Even if some burns reach high temperatures, incomplete combustion is inevitable at the beginnings and ends of the burns (Cook and Velis, 2021). Third, open burning often occurs close to where



people live, resulting in high exposure levels. The pollutants can be directly inhaled, distributed in the environment and subsequently ingested, or absorbed through skin.

Despite the global health crisis caused by uncontrolled solid waste open burning, the quantity of pollutant emissions is uncertain. Due to lack of data, household solid waste open burning emissions are not often included in regional, national, or global emission inventories (Wiedinmyer et al., 2014). Estimating household waste burning emissions faces two challenges: 1) it is difficult to estimate when, where, and how much burning occurs (activities); and 2) not many studies have systematically quantified representative open burning emission factors (EFs; i.e., amount of pollutant emitted per kg of fuel burned).

Several approaches have been used to derive EFs. The Intergovernmental Panel on Climate Change (Ipcc, 2006) calculates $CO_2$ EFs from carbon content in several categories of solid waste fuels. Bond et al. (2004) used a single $PM_{10}$ emission factor value of 30 g kg$^{-1}$ to represent all solid waste open burning when establishing a global inventory of black and organic carbon emissions. The U.S. Environmental Protection Agency (U.S. EPA) tested solid waste emissions when compiling and validating EFs in its AP-42 Compilation of Air Emissions Factors (U.S. Epa, 1992; Gerstle and Kemnitz, 1967; Lemieux, 1997, 1998).

However, many of the fuels do not represent modern waste materials and the applied measurement technologies are outdated. Other studies acquired laboratory emissions for several waste categories, such as shredded tires, plastic bags, and mixed garbage (Stockwell, 2016; Yokelson et al., 2013). Several field measurements were conducted in Nepal (Stockwell et al., 2016; Jayarathne et al., 2018). However, particle emissions were not often measured in these studies. While EFs for biomass burning are available, data for other waste categories, particularly those in Africa, are scant (Rabaji, 2019; Kwatala et al., 2019).

Developing more reliable EFs that represent open burning conditions has been identified as a research priority to reduce harm from solid waste open burning (Cook and Velis, 2021).

     To reduce emissions and improve air quality in surrounding communities near its facilities, SASOL, a petrochemical and energy company in South Africa, is implementing a Waste Collection Interventions (WCI) program to assist the Zamdela local community in MSW collection and disposal in landfills to minimize open burning. To improve emission inventories, this study

conducted comprehensive laboratory combustion experiments to determine household solid waste burning emissions. EFs for criterial pollutants from smoldering and flaming phases as well as the entire combustion process are reported for ten waste materials representing commonly disposed of in South Africa.

## 2 Method

### 2.1 Waste Materials

Mass distributions of common waste material categories that are burned in South Africa townships were obtained from SASOL's WCI program. As shown in Fig. 1, vegetation had the highest weight percent (33.3%), followed by plastics (20%) and paper (19.5%). Examples of major waste categories included in this study are illustrated in Supplemental Figure S1. Due to difficulties in preserving and importing food discards (organics) and vegetation, local substitutes (Nevada, USA) were used. Food waste was represented by a mixture of bread, potato and banana peels, lettuce, cucumbers, and tomatoes (Cronje et al,



2018). Vegetation samples were collected in Nevada to represent similar species in South Africa, including basin wild rye, Sandberg bluegrass, crested wheat grass, red willows, and creeping wild rye, typical of African bunch grasses, African sumac, and crab grass. EFs for glass, metals, and ceramics were not separately measured as they do not combust or degrade at open burning temperatures. However, to simulate their potential effects on combustion, these discards were included in the laboratory testing with combined waste materials. Ten types of waste categories/conditions were tested: 1) paper; 2) leather/rubber; 3) textile; 4) plastic bottles and food containers (hard plastics); 5) plastic bags (soft plastics); 6) dry vegetation (0% moisture content); 7) natural vegetation (20% moisture content); 8) damp vegetation (50% moisture content); 9) food discards; and 10) combined materials. The combined materials were mixtures of the other categories based on their mass fractions in Fig. 1. Each category was tested at least three times, except that the vegetations with 20% and 50% moisture content were each tested twice.

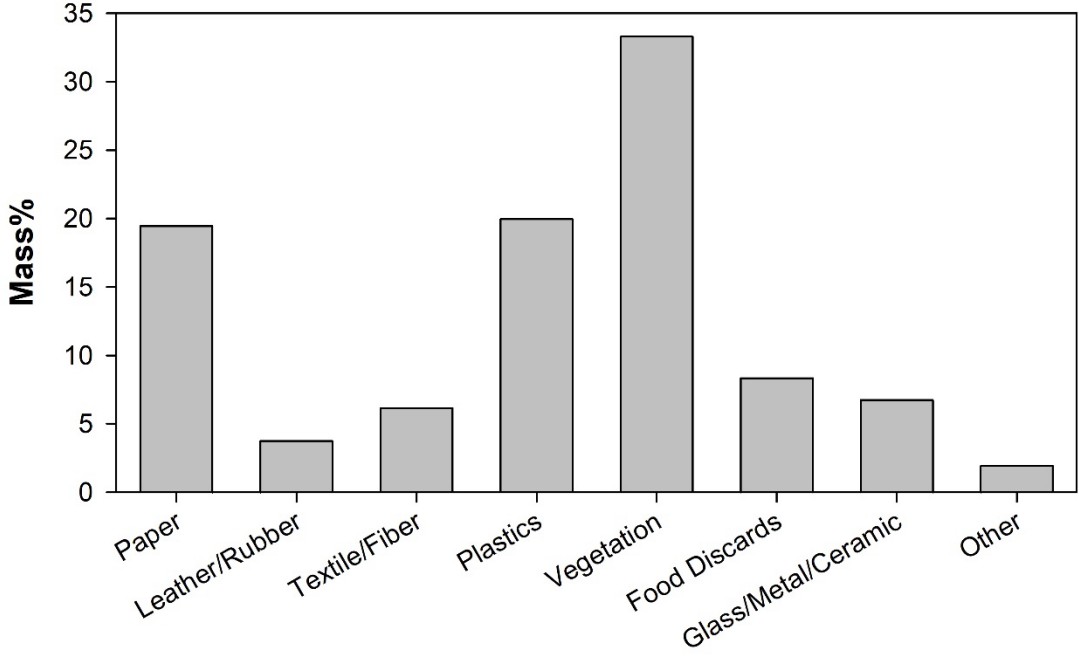

**Figure 1: Mass fraction of municipal solid waste categories collected by Sasol's Waste Collection Interventions (WCI) program in Zamdela, South Africa.**

Because fuel moisture content affects combustion behavior and emissions (Rein et al., 2008; Chen et al., 2010), the moisture contents were measured right after field collection, ranging 0.5–35% (Table S1). To account for moisture changes during shipping and storage, all materials (except food discards) were oven dried at 90 °C for 24 hours. A calculated amount of distilled deionized water (DDW) was then added to the dried materials to achieve the natural moisture levels shown in Table S1. These moisturized materials were sealed in airtight bags to equilibrate for at least 24 hours before testing. Fresh food



discards were tested without drying/re-moisturizing to avoid irreversible changes. The moisture content for the combined waste was calculated as the sum of the mass-weighed moisture content in individual waste category.

Table S2 shows the major elemental compositions (i.e., carbon [C], hydrogen [H], nitrogen [N], sulfur [S], and oxygen [O]) of the waste materials measured by an elemental analyzer (Model Flash EA1112, Thermo Scientific). Plastic bags (84%) and plastic bottles (64%) have higher carbon contents than other materials (33–48%). The carbon content is used for the fuel-based EF calculation. These C% contents fall within the IPCC (2006) range for all materials except the leather/rubber category: 33% (this study) vs. 67% (Ipcc, 2006). The single leather/rubber piece (a car floor mat) measured in this study may not be

representative of all such materials available elsewhere. Unlike other waste categories, IPCC (2006) does not give a range of C% for leather/rubber, indicating a need for a wider range of testing for this category.

The textile category contained the highest nitrogen (8%) and sulfur (0.71%) contents, while most other materials yielded sulfur contents below the minimum detection limit. The paper category had the highest oxygen content (44%), followed by vegetation and food discards (41–42%). The lowest (~3%) oxygen was found for soft plastic bags.

After combustion, the ash was weighed to calculate its mass fraction related to the original dry material mass, ranging from 2% to 58% (Table S3). The C, H, N, and S content of the ash was also measured by the elemental analyzer, and the ash C% is used in the EF calculation.

## 2.2 Combustion Experiments

The experimental setup is shown in Fig. 2, similar to the ones used in previous studies (Chen et al., 2010; Chow et al.,

2019; Tian et al., 2015; Wang et al., 2019, 2020b). Key specifications for gas and particle measurement instruments are listed in Table S4. For each experiment, a small amount (0.5 – 20 g) of waste material was placed in a ceramic crucible inside a woodstove, then quickly heated to and maintained at 450 °C by a temperature-controlled heater to simulate large scale open burning. The heater accounts for open burning temperatures surrounding the fuel materials that could be much higher than those produced by laboratory fuels (Chen et al., 2010; Chow et al., 2019). Flammable waste materials (i.e., paper, textile,

plastic bag, dry and natural moist vegetations, and combined wastes) were ignited by an electric heat gun or a butane lighter. Smoldering emissions were measured for non-flammable materials (i.e., leather/rubber, plastic bottle, damp vegetation, and food discards) until the pollutant concentrations returned to baselines. Elapsed time varied from 1000 to 4000 s for each burn, with typical run times of 30 min per sample. An exhaust fan drew fresh air through the stove inlet and vented the smoke above the roof via the stack. Temperature and relative humidity (RH) of the exhaust air were monitored by a hygrometer (Model

HH314A, Omega). A web camara inside the stove recorded the combustion process.

During combustion, major fuel components of C, H, N, and S are oxidized to generate carbon dioxide ($CO_2$), carbon monoxide (CO), water ($H_2O$), oxides of nitrogen ($NO_x$), sulfur dioxide ($SO_2$), volatile organic compounds (VOCs), and particulate matter (PM) (Akagi et al., 2011). The air sample was extracted from the stack through a sampling line and directed to a suite of gas and particle analyzers (Table S4). $CO_2$ was measured by a $CO_2$ analyzer (Model 840A; Li-Cor). CO was

measured by a CO analyzer (Model 48i, ThermoFisher Scientific), which is designated as a federal equivalent method (FEM)



by the U.S. Environmental Protection Agency (U.S. EPA). CO and $CO_2$ concentrations were used to calculate the modified combustion efficiency (MCE) and fuel-based EFs. $SO_2$ was measured by a FEM $SO_2$ analyzer (Model 43i, ThermoFisher Scientific). Nitric oxide (NO), nitrogen dioxide ($NO_2$), and $NO_x$ were measured by a FEM $NO/NO_2/NO_x$ analyzer (Model APNA-360, Horiba Ltd). An emission analyzer (Model 350 XL, Testo Inc.) provided redundant measurements of $CO_2$, CO,

$SO_2$, NO, and $NO_2$, in order to accommodate high concentrations in the event that the FEM analyzers were saturated. In addition, the Testo also measured oxygen ($O_2$) temperature (T) and pressure (P). Size segregated PM mass concentrations were acquired every second by an aerosol monitor (Model DustTrak DRX, TSI Inc.) in five size fractions (i.e., $PM_1$, $PM_{2.5}$, $PM_4$, $PM_{10}$, and $PM_{15}$) (Wang et al., 2009). Gas and particle analyzers were calibrated before and after experiments. All analyzer responses were quality checked to ensure readings were within their measurement ranges.

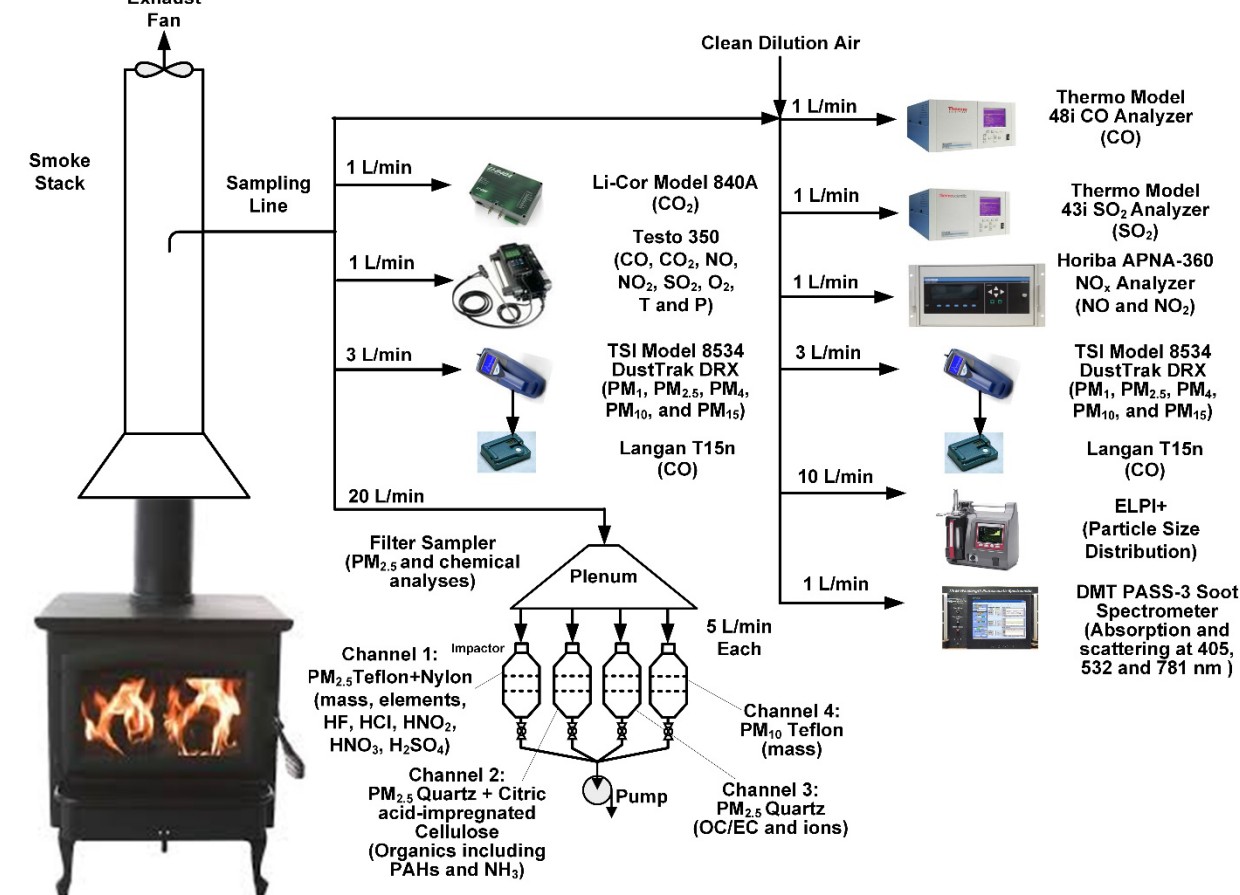

**Figure 2: Experimental setup for solid waste combustion.**

$PM_{2.5}$ and $PM_{10}$ samples were collected on Teflon-membrane and quartz-fiber filters. The gravimetric mass concentrations were used to calibrate the real-time mass concentrations by the DRX. Organic and elemental carbon (OC and EC) were

analyzed from the quartz-fiber filters using the DRI Model 2015 Carbon Analyzer following the IMPROVE_A protocol (Chow



et al., 2007; Chen et al., 2015). The filters were also analyzed for detailed chemical composition of PM$_{2.5}$, which will be reported in a future publication.

## 2.3 Data Analysis

Data from real-time gas and particle analyzers were assembled and mapped to a common time stamp with one-second
time resolution. Time series of gas and particle concentrations were aligned to account for their different transport and response times. Calibration factors were applied to each analyzer. Modified combustion efficiency (MCE) was calculated as

$$MCE = \frac{\Delta CO2}{\Delta CO2 + \Delta CO} \qquad (1)$$

where $\Delta CO_2$ and $\Delta CO$ are CO$_2$ and CO concentrations above background. MCE provides a real-time indicator of the combustion phase (i.e., MCE $\geq$ ~0.9 for flaming and MCE < 0.9 for smoldering) (Reid et al., 2005; Yokelson et al., 1996;
Wang et al., 2020a).

Fuel-based emission factors ($EF_{p,i}$) were calculated based on carbon mass balance techniques as (Wang et al., 2019; Chen et al., 2007; Moosmüller et al., 2003):

$$EF_{p,i} = \left( CMF_{fuel} - \frac{m_{ash}}{m_{fuel}} CMF_{ash} \right) \frac{C_p}{C_{CO_2}\left(\frac{M_c}{M_{CO_2}}\right) + C_{CO}\left(\frac{M_c}{M_{CO}}\right) + C_{PM}} \times 1000 \qquad (2)$$

where $EF_{p,i}$ is the emission factor of pollutant $p$ from waste material $i$ in g per kg of fuel. $CMF_{fuel}$ is the carbon mass fraction
of the fuel in g carbon per g of fuel (Table S2), and $CMF_{ash}$ is the carbon mass fraction of the ash in g carbon per g of ash (Table S3). $m_{ash}$ and $m_{fuel}$ are the mass of ash and fuel in g, respectively. $C_p$ is the plume concentration of pollutant $p$ in g m$^{-3}$; and $C_{CO}$ and $C_{CO2}$ are the concentrations of CO$_2$ and CO in g m$^{-3}$, respectively. $C_{PM}$ is the total caron (TC = OC + EC) concentration in PM$_{10}$ in g m$^{-3}$. $M_C$, $M_{CO2}$ and $M_{Co}$ are the atomic or molecular weights of carbon, CO$_2$, and CO in g per mole, respectively. The factor of 1000 converts mass from kilograms to grams. Eq. (2) assumes that the carbon mass in emissions
other than CO$_2$, CO, and PM$_{10}$ is negligible, which is a reasonable assumption for such burns. However, it is recognized that some carbon will be emitted as methane (CH$_4$) and VOCs, causing the EFs determined by Eq. (2) slightly overestimated. For waste materials that had both flaming and smoldering combustions, the split points between the two phases were determined from the burn video recording and MCE. $EF_{p,i}$ for flaming, smoldering, and the entire burning process were calculated. Means and standard deviations of $EF_{p,i}$ for each waste category and/or burn condition were calculated from repeated tests.

## 3 Results and Discussion

### 3.1 Evolution of Air Pollutants during Combustion

Time series plots of criteria pollutant concentrations, along with photographs of the waste materials, ash, and sample filters for each waste category are presented in Supplementary Section S3 to provide more details on the emission evolution, flaming



vs. smoldering phases, ash contents, and potential light absorption properties for each fuel. Results for plastic bottles and bags

are presented below to illustrate experimental findings from smoldering- and flaming-dominated combustions, respectively.

Trial burns with ~5 g of mixed plastic bottles generated very high PM concentrations that clogged filters and overloaded real-time particle sampling instruments. The final tests for this fuel utilized 0.5 g of material moisturized to 0.54% water content (Figure S13a). As shown in Fig. 3, smoldering started ~100 s after initial heating with low $CO_2$ and CO concentrations. PM emissions were the highest among all the waste materials, likely formed from re-condensation of evaporated plastic

molecules. The MCE was only ~0.6 during most of the burn, indicating low combustion efficiencies. $NO_x$ concentrations were close to background levels due to the low combustion temperatures and low nitrogen content of the fuel (Table S2).

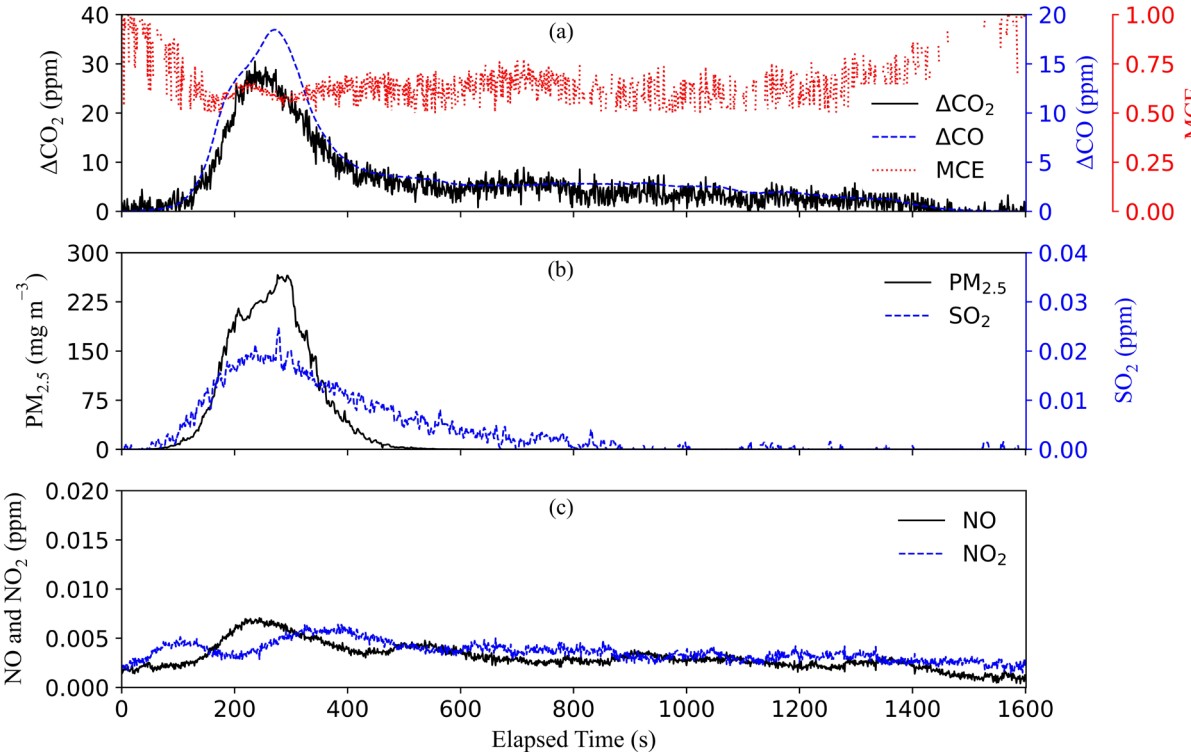

**Figure 3: Time series of emissions during a plastic bottle burning experiment.**

For the plastic bag experiment, 5 g of mixed soft plastic bags (Figure S16a) were prepared with 0.54% moisture content.

Flaming started at ~150 s after ignition, causing all pollutant concentrations to increase (Fig. 4). In contrast to the smoldering-only plastic bottle combustion, flaming dominated the soft plastics combustion, generating ~20 times higher $CO_2$ and CO concentrations. The shaded area in Fig. 4 shows the period during which flame was visible from the video camera. The MCE was high (> 0.94) during most parts of burn, indicating high combustion efficiencies. Plastic bags produced the highest $CO_2$ and the lowest CO EFs among all test materials, consistent with their high C and H content (Table S2). Only a small amount

of ash (3.4%) remained after combustion (Figure S16b).



Among the ten waste types, paper, textile, soft plastic bags, vegetations with dry and natural moisture contents, and combined waste had both flaming and smoldering phases. Leather/rubber, plastic bottles, damp vegetation, and food discards only smoldered. Ash residues were the highest for rubber (~58%) (Table S3), consistent with its high fraction of elements other than C, H, N, S, and O (Table S2). Similar flaming-dominated burns were found for vegetations with 0% and 20% moisture content (Figure S20 and Figure S21). In contrast to the smoldering dominated 50% moist vegetation that charred but did not flame (Figure S22). The mean MCEs for 0%, 20%, and 50% moisture content vegetations were ~0.92, 0.9, and 0.8, respectively, signifying the role of the moisture in the combustion efficiency (Chen et al., 2010).

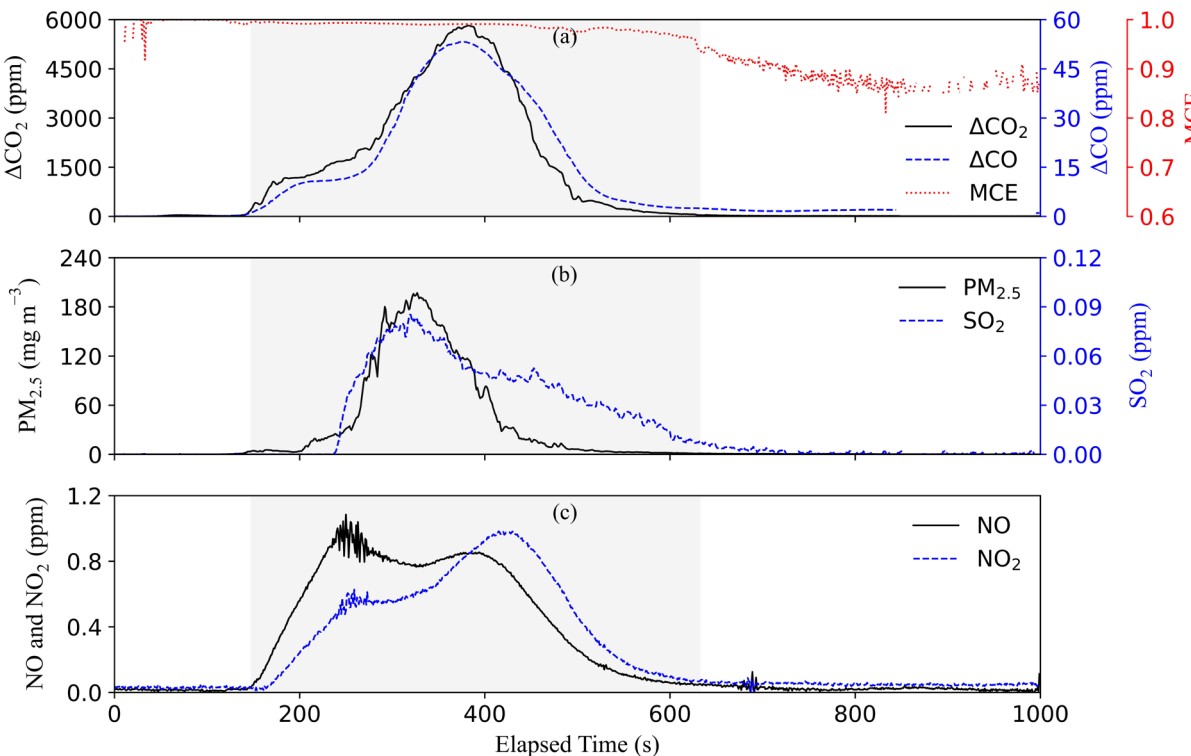

**Figure 4: Time series of emissions during a plastic bag burning experiment. The shaded areas indicate flaming stage.**

## 3.2 PM$_{2.5}$, PM$_{10}$, and Particulate Carbon

Figure 5 shows high correlations ($R^2 = 1$) between PM$_{2.5}$ and PM$_{10}$ mass for 30 sample sets. The linear regression slopes indicate that PM$_{2.5}$ constituted ~93% PM$_{10}$, consistent with findings for other combustion emissions.

Since the DRX measures PM concentration based on light scattering and its conversion from the scattering signal to mass concentration depends on particle refractive index, density, and size distribution, the DRX concentrations need to be calibrated with gravimetric concentrations (Wang et al., 2009). The mean DRX and gravimetric PM$_{2.5}$ and PM$_{10}$ mass concentrations are highly correlated with $R^2$ of 0.95–0.96 (Figure S2). The DRX measured mass concentrations with standard calibration were about twice of those by gravimetry (slopes of 1.88 for PM$_{2.5}$ and 1.82 for PM$_{10}$). The DRX had an internal custom photometric

calibration factor (PCF) of 1.0 and size calibration factor (SCF) of 1.7. The higher DRX reported concentrations are expected because the standard calibration uses Arizona Road Dust (ARD) with a density of 2.65 g cm$^{-3}$ (Wang et al., 2009) while the

major compositions of the combustion particles are OC and EC, which have lower densities (~1.8 and 1.1–1.4 g cm$^{-3}$, respectively) (Schmid et al., 2009). The DRX concentrations are normalized to the gravimetric PM$_{2.5}$ and PM$_{10}$ concentrations for EF calculations.

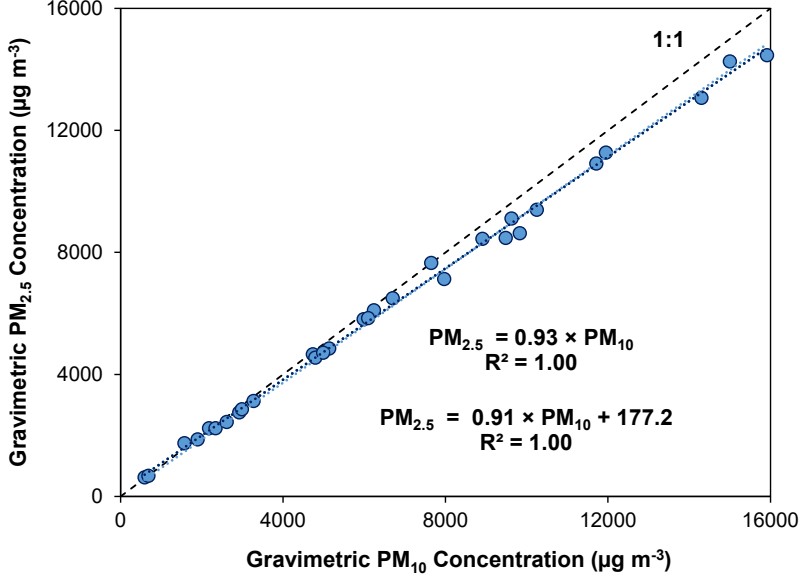

**Figure 5: Comparison of PM$_{2.5}$ and PM$_{10}$ mass concentrations measured from the Teflon-membrane filters.**

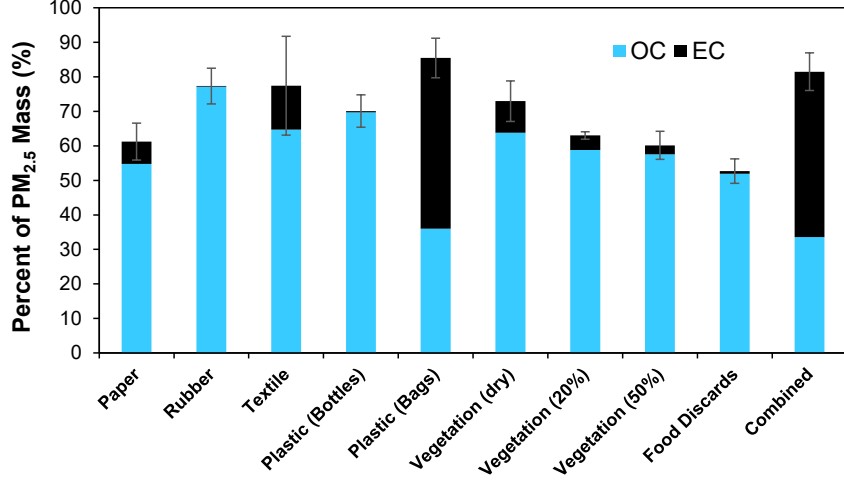

**Figure 6: Mass percent of organic carbon (OC) and elemental carbon (EC) in PM$_{2.5}$. The error bar indicates the uncertainty of total carbon (TC = OC + EC), calculated as the larger of the analytical uncertainty and standard deviation of multiple runs.**





Carbon is the most abundant $PM_{2.5}$ component. As shown in Fig. 6, TC contributed 70±11% (ranging 51–94%) of $PM_{2.5}$ mass emissions, with higher OC found in smoldering dominated materials (i.e., rubber, plastic bottles, damp vegetation, and

food discards). The EC fraction increased during flaming combustion, particularly for plastic bags and combined materials. Since $PM_{10}$ is only ~7% higher than $PM_{2.5}$ (Fig. 5), it is reasonable to assume that $PM_{2.5}$ and $PM_{10}$ have comparable TC fractions. The $C_{PM}$ in Eq. (2) was calculated from the TC fraction in $PM_{2.5}$ (Fig. 6) multiplied by the $PM_{10}$ mass concentration.

The properties and abundances of OC and EC affect the optical properties of PM emissions. Photographs of sample filters in Section S3 show that particles from flaming-dominated combustion of textiles, plastic bags, and combined materials have

grey to black coloration due to high EC abundances. Some OC-abundant filters do not show colors (e.g., rubber and plastic bottles) or show yellow/brown colors (e.g., paper, damped vegetation, and food discards), suggesting the presence of brown carbon (Andreae and Gelencsér, 2006; Chen et al., 2021). Quantitative analysis of particle optical properties will be reported in a separate publication.

### 3.3 Emission Factors (EF) for Criteria Pollutants

The fractions of consumed waste materials and emissions during flaming and smoldering phases for each category are listed in Table 1. Mean EFs for criteria pollutants are reported in Table 2 for flaming and smoldering phases, as well as for the entire combustion process. The relative standard deviations (RSD) of total EFs from multiple tests were within 50% of the mean, confirming reproducibility. Except for plastic bags that have high EFs due to high carbon fuel content, $CO_2$ and CO EFs are relatively consistent for materials that have both flaming and smoldering phases (i.e., paper, textile, dry and natural

vegetation, and combined waste), with RSD of 3% and 25%, respectively, in part due to similar fuel carbon contents as shown in Table S2 (RSD = 6%). Several exceptions with high RSD (e.g., $NO_x$ for textile and plastic bottles) were due to fuel material heterogeneity or low emission levels. The RSD for the flaming phases and smoldering phases were higher than those for the entire burns due to a somewhat subjective split between the two phases. Table 3 compares EFs from this study with those reported in the literature for similar fuel materials.

For paper, most of the fuel (76%) was consumed in the flaming phase (Table 1), consistent with elevated $CO_2$ concentrations (Figure S4). Approximately 65–85% of pollutants were emitted in the flaming phase except for CO, which was emitted about equally in both phases. EFs for CO in the smoldering phase were ~4 times of those in flaming phase. EFs for paper combustion are scarce in the literature (Table 3). The EFs for $PM_{2.5}$ and $PM_{10}$ reported by Park et al. (2013) were an order of magnitude lower than those from this study. Paper briquettes used in the Marshall Islands (Thai et al., 2016; Xiu et

al., 2018) likely have different combustion behaviors compared to the open burning of loose paper; therefore, and the EFs are not considered to be comparable.

The car floor mat rubber sample only smoldered without flaming, leading to low $CO_2$ and high PM EFs (Table 2). A large fraction (58%) of material was unburned as ash with a 13% carbon content (Table S3). Field and laboratory studies of tire burning emissions (Ryan, 1989; Downard et al., 2015; Stockwell, 2016) report higher EFs than those found here for most

pollutants, but $PM_{10}$ EFs are similar.



Textile burning consumed 78% of the mass and emitted 60–90% pollutants in the flaming phase except for ~20% more CO emissions in the smoldering phase (Table 1). While EFs for $CO_2$ and $SO_2$ were higher in the flaming phase, EFs for CO and PM were higher in the smoldering phase (Table 2). Textile burning had the highest EFs for $NO_x$ and $SO_2$ among all tested materials, consistent with higher nitrogen and sulfur contents (Table S1). Wesolek and Kozlowski (2002) measured gas
emissions during thermal decomposition of natural and synthetic fabrics at 450, 550, and 750 °C. The EFs for $CO_2$ and CO from this study fall within the ranges of those reported for different fabrics (Table 3). EFs for $NO_x$ and $SO_2$ were higher in this study, likely due to differences in material compositions.

The plastic bottles only smoldered, yielding the lowest $CO_2$ EFs and among the highest CO and PM EFs (Table 2). Most fuel carbon was turned into PM and volatile organics (strong odor). In contrast, flaming dominated plastic bag combustion,
consuming ~99% of the fuel mass and contributing to over 90% of emissions (Table 1). Among all waste materials, plastic bags had the highest $CO_2$ EFs due to their high carbon content (Table S1) and high combustion efficiencies. Similar high efficiency combustion of plastic bags is reported by Stockwell (2016). Plastic bag EFs were in the same range as literature values. Note that the literature has a wide range of PM EFs, likely due to different plastic materials and burning conditions (Table 3).

The flaming phase for vegetations with 0% and 20% moisture content consumed ~70% of the fuel mass and emitted over 70% of pollutants, except that ~60–75% of the CO was emitted during smoldering (Table 1). The damp 50% moisture content vegetation emitted 26% less $CO_2$, but a factor of 3 and 30 higher CO and PM, respectively, as compared to the drier vegetations. Most of the published vegetation emissions lack information on moisture content. Some studies with fuels relevant to South Africa are compared in Table 3. The EFs are consistent with those of low moisture contents measured in this study. In
particular, EFs for $CO_2$, CO, and $SO_2$ derived here are in good agreement with those derived for Savanna vegetation (Akagi et al., 2011). The EFs for PM from damp vegetation burning were about one order of magnitude higher than literature values.

Food discards did not flame due to high moisture contents in fresh vegetables and fruits, resulting in lower EFs for $CO_2$ and higher EFs for CO and PM (Table 2). Food discards are often included in municipal/household waste, but no separate EFs for food discard burning have been found in the literature.

Flaming-dominated combustion of the combined materials consumed 81% of the fuel mass and emitted over 75% of the pollutants, except that 62% of the CO was emitted during smoldering (Table 1). Combined waste combustion was efficient and MCE for most of the burn period was higher than 0.90 (Figure S29). The EFs for combined waste fall within the EF ranges of the individual waste categories, but with lower EFs for PM (Table 2). Considering the wide variety of waste materials and burn practices, EFs are expected to vary over a wide range. Interestingly, as shown in Table 3, with the exception of an old
(1967) test in the USA (U.S. Epa, 1992; Gerstle and Kemnitz, 1967) with a "below average" data quality rating, most recent studies show reasonable consistency in EFs. EFs for $CO_2$ and CO from this study agree remarkably well with data suggested for global emission inventory development (Akagi et al., 2011; Reyna-Bensusan et al., 2018; Wiedinmyer et al., 2014).



**Table 1: Percentage of consumed fuel and emissions during flaming and smoldering phases.**

| Fuel | Burn Type | Relative Fraction of Fuel Burned and Emissions in Flaming and Smoldering Phases (%) | | | | | | | | |
|------|-----------|------------------|-------------|-------------|-------------|-------------|-------------|-------------|-------------|-------------|
| | | Burned Fuel Mass | $CO_2$ | CO | NO | $NO_2$ | $NO_x$ | $SO_2$ | $PM_{2.5}$ | $PM_{10}$ |
| Paper | Flaming | 76 ± 8 | 77 ± 7 | 46 ± 18 | 72 ± 12 | 64 ± 16 | 68 ± 14 | 84 ± 5 | 69 ± 22 | 69 ± 22 |
| | Smoldering | 24 ± 8 | 23 ± 7 | 54 ± 18 | 28 ± 12 | 36 ± 16 | 32 ± 14 | 16 ± 5 | 31 ± 22 | 31 ± 22 |
| Leather/ Rubber | Flaming | No Flaming Phase | | | | | | | | |
| | Smoldering | 100 | | | | | | | | |
| Textile | Flaming | 78 ± 8 | 81 ± 6 | 41 ± 19 | 75 ± 19 | 76 ± 18 | 75 ± 19 | 90 ± 2 | 61 ± 23 | 60 ± 23 |
| | Smoldering | 22 ± 8 | 19 ± 6 | 59 ± 19 | 25 ± 19 | 24 ± 18 | 25 ± 19 | 10 ± 2 | 39 ± 23 | 40 ± 23 |
| Plastic Bottles | Flaming | No Flaming Phase | | | | | | | | |
| | Smoldering | 100 | | | | | | | | |
| Plastic Bags | Flaming | 99 ± 0 | 99 ± 0 | 93 ± 2 | 96 ± 2 | 93 ± 2 | 94 ± 2 | 96 ± 2 | 97 ± 3 | 97 ± 3 |
| | Smoldering | 1 ± 0 | 1 ± 0 | 7 ± 2 | 4 ± 2 | 7 ± 2 | 6 ± 2 | 4 ± 2 | 3 ± 3 | 3 ± 3 |
| Vegetation (0% mc*) | Flaming | 72 ± 4 | 75 ± 4 | 26 ± 1 | 80 ± 3 | 77 ± 5 | 80 ± 3 | 94 ± 1 | 87 ± 8 | 87 ± 8 |
| | Smoldering | 28 ± 4 | 25 ± 4 | 74 ± 1 | 20 ± 3 | 23 ± 5 | 20 ± 3 | 6 ± 1 | 13 ± 8 | 13 ± 8 |
| Vegetation (20% mc*) | Flaming | 70 ± 3 | 72 ± 1 | 43 ± 18 | 77 ± 0 | 81 ± 1 | 79 ± 0 | 94 ± 2 | 91 ± 4 | 91 ± 4 |
| | Smoldering | 30 ± 3 | 28 ± 1 | 57 ± 18 | 23 ± 0 | 19 ± 1 | 21 ± 0 | 6 ± 2 | 9 ± 4 | 9 ± 4 |
| Vegetation (50% mc*) | Flaming | No Flaming Phase | | | | | | | | |
| | Smoldering | 100 | | | | | | | | |
| Food Discards | Flaming | No Flaming Phase | | | | | | | | |
| | Smoldering | 100 | | | | | | | | |
| Combined | Flaming | 81 ± 0 | 83 ± 1 | 38 ± 2 | 75 ± 2 | 83 ± 3 | 78 ± 2 | 97 ± 1 | 82 ± 8 | 82 ± 8 |
| | Smoldering | 19 ± 0 | 17 ± 1 | 62 ± 2 | 25 ± 2 | 17 ± 3 | 22 ± 2 | 3 ± 1 | 18 ± 8 | 18 ± 8 |

*mc: fuel moisture content


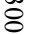


**Table 2: Measured emission factors (mean ± standard deviation) for waste materials tested in this study.**

| Fuel | Burn Type | Mean MCE | Emission Factor (g kg⁻¹ fuel) | | | | | | | |
| --- | --- | --- | --- | --- | --- | --- | --- | --- | --- | --- |
| | | | $CO_2$ | CO | NO (as $NO_2$) | $NO_2$ | $NO_x$ (as $NO_2$) | $SO_2$ | $PM_{2.5}$ | $PM_{10}$ |
| Paper | Flaming | 0.96 ± 0.03 | 1530 ± 24 | 26.2 ± 6.9 | 0.58 ± 0.04 | 0.42 ± 0.15 | 1.00 ± 0.15 | 0.68 ± 0.58 | 12.05 ± 3.28 | 12.19 ± 3.70 |
| | Smoldering | 0.87 ± 0.04 | 1406 ± 22 | 101.2 ± 13.3 | 0.81 ± 0.51 | 0.86 ± 0.53 | 1.66 ± 1.00 | 0.33 ± 0.08 | 15.21 ± 6.96 | 15.16 ± 6.67 |
| | Total | 0.90 ± 0.02 | 1498 ± 7 | 44.9 ± 3.2 | 0.63 ± 0.16 | 0.52 ± 0.19 | 1.14 ± 0.31 | 0.57 ± 0.41 | 13.31 ± 0.77 | 13.42 ± 1.21 |
| Rubber | Flaming | 0.92 ± 0.02 | 456 ± 41 | 28.1 ± 3.9 | 0.31 ± 0.15 | 2.75 ± 4.44 | 3.06 ± 4.59 | 0.16 ± 0.04 | 141.34 ± 23.01 | 153.19 ± 20.26 |
| | Smoldering | 0.87 ± 0.03 | 1227 ± 59 | 149.5 ± 34.5 | 11.57 ± 8.73 | 1.19 ± 0.53 | 12.76 ± 9.87 | 1.68 ± 0.45 | 75.56 ± 15.33 | 87.55 ± 24.71 |
| Textile | Flaming | 0.97 ± 0.01 | 1540 ± 129 | 27.3 ± 8.9 | 9.53 ± 1.95 | 1.17 ± 0.19 | 10.70 ± 5.58 | 4.43 ± 2.12 | 37.20 ± 22.65 | 42.78 ± 31.32 |
| | Smoldering | 0.86 ± 0.03 | 1227 ± 59 | 149.5 ± 34.5 | 11.57 ± 8.73 | 1.19 ± 0.53 | 12.76 ± 9.87 | 1.68 ± 0.45 | 75.56 ± 15.33 | 87.55 ± 24.71 |
| | Total | 0.87 ± 0.03 | 1467 ± 104 | 54.9 ± 7.4 | 10.37 ± 3.72 | 1.21 ± 0.15 | 11.58 ± 6.57 | 3.72 ± 1.48 | 47.04 ± 16.83 | 53.95 ± 26.96 |
| Plastic Bottles | Flaming | 0.56 ± 0.05 | 182 ± 42 | 90.4 ± 10.6 | 0.22 ± 0.26 | 0.12 ± 0.08 | 0.35 ± 0.34 | 0.22 ± 0.02 | 651.00 ± 38.45 | 722.47 ± 17.98 |
| | Smoldering | 0.56 ± 0.05 | 182 ± 42 | 90.4 ± 10.6 | 0.22 ± 0.26 | 0.12 ± 0.08 | 0.35 ± 0.34 | 0.22 ± 0.02 | 651.00 ± 38.45 | 722.47 ± 17.98 |
| | Total | 0.56 ± 0.05 | 182 ± 42 | 90.4 ± 10.6 | 0.22 ± 0.26 | 0.12 ± 0.08 | 0.35 ± 0.34 | 0.22 ± 0.02 | 651.00 ± 38.45 | 722.47 ± 17.98 |
| Plastic Bags | Flaming | 0.98 ± 0.00 | 2938 ± 26 | 21.0 ± 5.1 | 0.70 ± 0.17 | 0.72 ± 0.04 | 1.42 ± 0.14 | 0.08 ± 0.01 | 33.48 ± 9.22 | 36.01 ± 9.62 |
| | Smoldering | 0.89 ± 0.01 | 2506 ± 247 | 183.9 ± 13.7 | 3.74 ± 0.82 | 6.87 ± 2.62 | 10.61 ± 3.15 | 0.36 ± 0.17 | 85.75 ± 76.56 | 89.47 ± 76.47 |
| | Total | 0.94 ± 0.01 | 2934 ± 24 | 22.4 ± 5.4 | 0.72 ± 0.17 | 0.77 ± 0.06 | 1.50 ± 0.12 | 0.08 ± 0.01 | 34.00 ± 8.55 | 36.55 ± 8.88 |
| Vegetation (0% mc[a]) | Flaming | 0.97 ± 0.01 | 1573 ± 11 | 21.0 ± 3.6 | 2.94 ± 0.42 | 0.40 ± 0.15 | 3.34 ± 0.21 | 0.72 ± 0.14 | 3.80 ± 1.07 | 3.60 ± 0.83 |
| | Smoldering | 0.84 ± 0.02 | 1366 ± 18 | 156.2 ± 13.6 | 1.87 ± 0.16 | 0.29 ± 0.03 | 2.17 ± 0.12 | 0.12 ± 0.02 | 1.70 ± 1.68 | 1.57 ± 1.48 |
| | Total | 0.88 ± 0.01 | 1515 ± 12 | 58.5 ± 4.8 | 2.64 ± 0.32 | 0.37 ± 0.12 | 3.01 ± 0.11 | 0.54 ± 0.08 | 3.20 ± 1.25 | 3.02 ± 1.01 |
| Vegetation (20% mc[a]) | Flaming | 0.93 ± 0.04 | 1549 ± 14 | 34.7 ± 8.1 | 2.42 ± 0.13 | 0.74 ± 0.12 | 3.16 ± 0.24 | 0.76 ± 0.10 | 5.40 ± 1.00 | 5.56 ± 1.14 |
| | Smoldering | 0.87 ± 0.02 | 1390 ± 7 | 135.5 ± 15.2 | 1.43 ± 0.08 | 0.47 ± 0.09 | 1.90 ± 0.01 | 0.20 ± 0.08 | 5.88 ± 7.27 | 6.18 ± 7.68 |
| | Total | 0.91 ± 0.03 | 1505 ± 1 | 63.9 ± 3.3 | 2.17 ± 0.07 | 0.64 ± 0.07 | 2.82 ± 0.13 | 0.56 ± 0.07 | 4.80 ± 1.98 | 4.97 ± 2.16 |
| Vegetation (50% mc[a]) | Flaming | 0.79 ± 0.00 | 1124 ± 0 | 183.6 ± 0.7 | 1.64 ± 0.15 | 0.25 ± 0.04 | 1.88 ± 0.19 | 0.28 ± 0.05 | 87.57 ± 6.83 | 92.66 ± 7.24 |
| | Smoldering | 0.79 ± 0.00 | 1124 ± 0 | 183.6 ± 0.7 | 1.64 ± 0.15 | 0.25 ± 0.04 | 1.88 ± 0.19 | 0.28 ± 0.05 | 87.57 ± 6.83 | 92.66 ± 7.24 |
| | Total | 0.79 ± 0.00 | 1124 ± 0 | 183.6 ± 0.7 | 1.64 ± 0.15 | 0.25 ± 0.04 | 1.88 ± 0.19 | 0.28 ± 0.05 | 87.57 ± 6.83 | 92.66 ± 7.24 |
| Food | Flaming | 0.89 ± 0.01 | 955 ± 30 | 76.1 ± 7.6 | 1.71 ± 0.34 | 0.27 ± 0.01 | 1.98 ± 0.34 | 0.16 ± 0.02 | 82.97 ± 18.36 | 87.23 ± 20.76 |
| | Smoldering | 0.89 ± 0.01 | 955 ± 30 | 76.1 ± 7.6 | 1.71 ± 0.34 | 0.27 ± 0.01 | 1.98 ± 0.34 | 0.16 ± 0.02 | 82.97 ± 18.36 | 87.23 ± 20.76 |
| | Total | 0.89 ± 0.01 | 955 ± 30 | 76.1 ± 7.6 | 1.71 ± 0.34 | 0.27 ± 0.01 | 1.98 ± 0.34 | 0.16 ± 0.02 | 82.97 ± 18.36 | 87.23 ± 20.76 |
| Combined | Flaming | 0.98 ± 0.00 | 1443 ± 8 | 14.9 ± 0.7 | 1.66 ± 0.14 | 0.63 ± 0.03 | 2.29 ± 0.16 | 1.13 ± 0.15 | 6.94 ± 2.32 | 7.34 ± 2.36 |
| | Smoldering | 0.88 ± 0.02 | 1302 ± 28 | 105.1 ± 11.0 | 2.40 ± 0.19 | 0.55 ± 0.09 | 2.95 ± 0.26 | 0.17 ± 0.06 | 6.55 ± 3.01 | 6.95 ± 3.22 |
| | Total | 0.91 ± 0.01 | 1417 ± 8 | 31.6 ± 1.8 | 1.80 ± 0.11 | 0.61 ± 0.00 | 2.41 ± 0.11 | 0.95 ± 0.13 | 6.86 ± 2.08 | 7.26 ± 2.12 |

[a] mc: fuel moisture content



**Table 3: Comparison of emission factors from this study with those reported in the literature.**

| Ref. | Region | Fuel | \multicolumn Emission Factor (g kg⁻¹ fuel) | | | | | | Method |
|---|---|---|---|---|---|---|---|---|---|
| | | | $CO_2$ | CO | $NO_x$ (as $NO_2$) | $SO_2$ | $PM_{2.5}$ | $PM_{10}$ | |
| | | | | | *Paper* | | | | |
| **This study** | **South Africa** | **Paper** | **1498 ± 7** | **44.9 ± 3.2** | **1.14 ± 0.31** | **0.57 ± 0.41** | **13.31 ± 0.77** | **13.42 ± 1.21** | **Lab** |
| (Park et al., 2013) | South Korea | Paper | | | | | 0.6 (0.25–0.8) | 0.93 (0.73–1.13) | Lab |
| (Thai et al., 2016; Xiu et al., 2018) | Marshall Islands | Paper briquettes | | 112 | 5.7 | 2.0 | | | Lab |
| | | | | | *Leather/Rubber/Tires* | | | | |
| **This study** | **South Africa** | **Car floor mat** | **456 ± 41** | **28.1 ± 3.9** | **3.06 ± 4.59** | **0.16 ± 0.04** | **141.34 ± 23.01** | **153.19 ± 20.26** | **Lab** |
| (Ryan, 1989) | USA | Chunk tire | | | | | | 108–119 | Lab |
| (Ryan, 1989) | USA | Shredded tire | | | | | 119–179 | | Lab |
| (Downard et al., 2015) | USA | Shredded tires | | | | 7.1±8.3 | 5.35±5.39 | | Field |
| (Stockwell, 2016) | USA | Shredded tire | 2882±14 | 70.6±6.4 | 7.81 | 26.2±2.2 | | | Lab |
| | | | | | *Textile/fabric* | | | | |
| **This study** | **South Africa** | **Mixed fabrics** | **1467 ± 104** | **54.9 ± 7.4** | **11.58 ± 6.57** | **3.72 ± 1.48** | **47.04 ± 16.83** | **53.95 ± 26.96** | **Lab** |
| (Wesolek and Kozlowski, 2002) | Poland | Natural fabrics | 850–1300 | 50–215 | 0.15–0.43 | 0.1–1.1 | | | |
| (Wesolek and Kozlowski, 2002) | Poland | Synthetic fabrics | 1000–1750 | 21–139 | 0.1–0.33 | 0.06–0.07 | | | Lab |
| | | | | | *Plastics* | | | | |
| **This study** | **South Africa** | **Plastic bottles** | **182 ± 42** | **90.4 ± 10.6** | **0.35 ± 0.34** | **0.22 ± 0.02** | **651.00 ± 38.45** | **722.47 ± 17.98** | **Lab** |
| **This study** | **South Africa** | **Plastic bags** | **2934 ± 24** | **22.4 ± 5.4** | **1.50 ± 0.12** | **0.08 ± 0.01** | **34.00 ± 8.55** | **36.55 ± 8.88** | **Lab** |
| (Park et al., 2013) | South Korea | Plastics | | | | | 0.5 (0.1–0.85) | 1.5 (0.6–2.4) | Lab |
| (Lemieux et al., 2004; Oberacker et al., 1992) | USA | Agricultural plastic film | | | | | 5.7 | 5.7 | Lab |

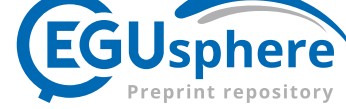

| Ref. | Region | Fuel | Emission Factor (g kg$^{-1}$ fuel) | | | | | | Method |
|---|---|---|---|---|---|---|---|---|---|
| | | | $CO_2$ | CO | $NO_x$ (as $NO_2$) | $SO_2$ | $PM_{2.5}$ | $PM_{10}$ | |
| (Stockwell et al., 2016; Jayarathne et al., 2018) | Nepal | Chip bags | 2249 | 15.9 | 4.30 | bdl[a] | 50±9 | | Field |
| (Stockwell et al., 2016; Jayarathne et al., 2018) | Nepal | Plastics | 2473–2695 | 16.6–62.2 | 5.31 | bdl[a] | 84±13 | | Field |
| (Stockwell, 2016) | USA | Plastic bag | 3127 | 11.7 | 2.69 | | | | Lab |
| Vegetation | | | | | | | | | |
| **This study** | **South Africa** | **Vegetation (0% mc[a])** | **1515 ± 12** | **58.5 ± 4.8** | **3.01 ± 0.11** | **0.54 ± 0.08** | **3.20 ± 1.25** | **3.02 ± 1.01** | **Lab** |
| **This study** | **South Africa** | **Vegetation (20% mc[a])** | **1505 ± 1** | **63.9 ± 3.3** | **2.82 ± 0.13** | **0.56 ± 0.07** | **4.80 ± 1.98** | **4.97 ± 2.16** | **Lab** |
| **This study** | **South Africa** | **Vegetation (50% mc[a])** | **1124 ± 0** | **183.6 ± 0.7** | **1.88 ± 0.19** | **0.28 ± 0.05** | **87.57 ± 6.83** | **92.66 ± 7.24** | **Lab** |
| (Christian et al., 2010) | Mexico | Barley stubble | 1602 | 118 | | | | | Field |
| (Akagi et al., 2011) | Africa | Savanna vegetation | 1686±38 | 63±17 | 6.0±1.2 | 0.48±0.27 | 7.17±3.42 | | Data synthesis |
| (Akagi et al., 2011) | Global | Crop residue | 1585±100 | 102±33 | 4.8±2.4 | | 6.26±2.36 | | |
| (Santiago-De La Rosa et al., 2018) | Mexico | Alfalfa | 1052±144 | 65.23±5.38 | | | 9.98 ± 0.71 | 11.11±0.91 | Lab |
| (Santiago-De La Rosa et al., 2018) | Mexico | Barley | 1693±84 | 33.31±2.33 | | | 1.19 ± 0.10 | 1.77±0.19 | Lab |
| (Santiago-De La Rosa et al., 2018) | Mexico | Bean | 1230±38 | 65.92±3.5 | | | 2.24 ± 0.19 | 2.75±0.18 | Lab |
| (Santiago-De La Rosa et al., 2018) | Mexico | Cotton | 1690±76 | 75.81±4.1 | | | 8.22 ± 0.54 | 13.37±1.9 | Lab |
| (Santiago-De La Rosa et al., 2018) | Mexico | Maize | 1748±81 | 34.61±2.04 | | | 2.70 ± 0.28 | 3.3±0.42 | Lab |
| (Santiago-De La Rosa et al., 2018) | Mexico | Rice | 1651±54 | 81.12±3.25 | | | 3.04 ± 0.24 | 4.95±0.52 | Lab |
| (Santiago-De La Rosa et al., 2018) | Mexico | Sorghum | 1851±58 | 155.71±4.77 | | | 11.30 ± 1.05 | 21.56±2.26 | Lab |
| (Santiago-De La Rosa et al., 2018) | Mexico | Wheat | 1812±103 | 28.85±1.79 | | | 2.54 ± 0.39 | 4.07±0.51 | Lab |
| (Yokelson et al., 2009) | Mexico | Crop residues | 1676±50 | 75.04±25.81 | 7.21±2.69 | | | | Field |
| (Yokelson et al., 2009) | Mexico | Deforestation | 1656±38 | 82.68±14.21 | 7.20±2.72 | | | | Field |




| Ref. | Region | Fuel | Emission Factor (g kg⁻¹ fuel) | | | | | | Method |
|---|---|---|---|---|---|---|---|---|---|
| | | | $CO_2$ | CO | $NO_x$ (as $NO_2$) | $SO_2$ | $PM_{2.5}$ | $PM_{10}$ | |
| (Ni et al., 2015) | China | Corn stalk | 1363±154 | 52.1±17.7 | | | 12.0±5.4 | | Lab |
| (Ni et al., 2015) | China | Rice straw | 1393±91 | 57.2±26.0 | | | 8.5±6.7 | | Lab |
| (Ni et al., 2015) | China | Wheat straw | 1311±181 | 47.9±13.5 | | | 11.4±4.9 | | Lab |
| Mixed household/municipal waste | | | | | | | | | |
| **This study** | **South Africa** | **Combined waste** | **1417±8** | **31.6±1.8** | **2.41±0.11** | **0.95±0.13** | **6.86±2.08** | **7.26±2.12** | **Lab** |
| (U.S. Epa, 1992; Gerstle and Kemnitz, 1967) | USA | Municipal refuse | 615 | 42 | 3 | 0.5 | | 8 (TSP)ᶜ | Lab |
| (Lemieux, 1997, 1998) | USA | Household waste (no recycle) | | | | | 14.8–20.07 | 16.23–21.28 | Lab |
| (Lemieux, 1997, 1998) | USA | Household waste (recycle) | | | | | 3.58–6.93 | 4.18–7.46 | Lab |
| (Christian et al., 2010) | Mexico | Landfill garbage | 1367±65 | 45.3±22.8 | | | 10.5±8.8 | | Field |
| (Park et al., 2013) | South Korea | Household solid waste | | | | | 0.78 (0.48–0.98) | 1.2 (0.3–1.9) | Lab |
| (Stockwell et al., 2016; Jayarathne et al., 2018) | Nepal | Mixed garbage | 1602±142 | 84.7±55.5 | 3.39±0.21 | bdlᵇ | 7.37±1.22 | | Field |
| (Jayarathne et al., 2018) | Nepal | Damp mixed garbage | | | | | 124±23 | 82±13 | Field |
| (Akagi et al., 2011; Reyna-Bensusan et al., 2018; Wiedinmyer et al., 2014) | Global | Mixed garbage | 1453±69 | 38±19 | 5.7±2.3 | 0.5 | 9.8±5.7 | | Data synthesis |
| (Yokelson et al., 2013) | USA | Mixed garbage | 1341 | 28.7 | 1.35 | 0.77 | 10.8 | 11.9 | Lab |
| (Stockwell, 2016) | USA | Mixed household refuse | 1793±28 | 31.5±6.9 | 1.57±0.41 | 0.897 | | | Lab |

ᵃmc: moisture content
ᵇbdl: below detection limit
ᶜTSP: total suspended particulate



Table 2 shows that $CO_2$ EFs are 10–25% higher for flaming compared to smoldering and are lowest for smoldering only combustions, while CO EFs are 4–9 times higher for smoldering than for flaming. Figure S3a and b show that overall, $CO_2$ increased with MCE while CO decreased with MCE, although there were large variations among fuel materials. Among the tested materials, textile has the highest nitrogen and sulfur contents, resulting in the highest EFs for $NO_x$ and $SO_2$. EFs for $NO_x$ are generally higher in the smoldering phase (except for vegetation), probably due to the time required for fuel nitrogen to be oxidized and released. Due to larger fuel influences, $NO_x$ emissions do not show a strong pattern as a function of MCE (Figure S3c). EFs for $SO_2$ are generally higher in the flaming phase (except for plastic bags). Figure S3d shows that EFs for $PM_{2.5}$ do not show a strong correlation with MCE. Over two-fold higher EFs are found in smoldering than flaming of textile and plastic bags, with less variations between the two phases for paper, vegetation, and combined materials.

**3.4 Effects of Ash and Particulate Carbon Content on EF Calculation**

Carbon contents in the ash or PM emissions (Eq. (2)) are rarely included in fuel-based EF calculations (Stockwell et al., 2016; Christian et al., 2010; Jayarathne et al., 2018; Wang et al., 2019; Chen et al., 2007). Their impacts are assumed to be negligible but have not been systematically evaluated. Table 4 demonstrates the importance of carbon in ash $\left(\frac{m_{ash}}{m_{fuel}} CMF_{ash}/CMF_{fuel}\right)$ and in PM $\left(C_{PM}/\left(C_{CO_2}\left(\frac{M_C}{M_{CO_2}}\right) + C_{CO}\left(\frac{M_C}{M_{CO}}\right) + C_{PM}\right)\right)$ in EF calculations using Eq. (2). Without including ash and/or PM carbon, changes in EFs are <5% for flaming dominated combustion of paper, plastic bags, vegetation with 0% and 20% moisture content, and combined materials. These fuels had <5% fuel carbon in ash and <5% emitted carbon in PM.

The consequences of not including ash or PM carbon are larger for smoldering fuels. Due to their high EFs of carbonaceous PM, the errors caused by not including PM carbon are over 10%. Rubber had the highest fuel carbon (22.6%) in the ash, and excluding ash in Eq. (2) results in a 29.1% overestimation of EFs. Rubber had 46.5% carbon emitted as TC in PM; excluding $C_{PM}$ causes an EF overestimation of 87%. If neither ash nor PM carbon is included, the EFs are overestimated by 141%. The hard plastic bottle EFs are also affected by carbon contents. Because of the very high EFs for carbonaceous PM and relatively low EFs for CO and $CO_2$, 85% of the carbon was emitted as PM. Not including $C_{PM}$ results in an EF overestimation of 577%; in addition, if ash carbon is not included, the EFs are overestimated by 623%.

This result shows that ash and PM carbon cannot be neglected in EF calculations, particularly for smoldering combustion with high carbon contents in ash and/or PM emissions. Carbon can also be emitted as gaseous hydrocarbons and excluding it in Eq. (2) may result in some overestimation of the EFs. While it is expected that the hydrocarbon carbon content is lower than that in CO and $CO_2$ in most cases, it may not be negligible when their emissions are high. Future studies should measure total hydrocarbons for more accurate EF determination.



**Table 4: Emission factor changes relative to Eq. (2) when the carbon in the PM (C$_{PM}$) or ash (CMF$_{ash}$) are not included.**

| Fuel | Fuel Carbon in Ash | Emitted Carbon in PM$_{10}$ | EF Changes relative to Eq. (2) | | |
|---|---|---|---|---|---|
| | | | With Ash Without C$_{PM}$ | Without Ash With C$_{PM}$ | Without Ash Without C$_{PM}$ |
| Paper | 1.1 ± 0.3% | 1.9 ± 0.3% | 1.9% | 1.1% | 3.1% |
| Rubber | 22.6 ± 1.0% | 46.5 ± 4.5% | 87.0% | 29.1% | 141.4% |
| Textile | 2.1 ± 0.4% | 9.4 ± 6.6% | 10.4% | 2.2% | 12.8% |
| Plastic Bottle | 6.4 ± 3.8% | 85.2 ± 1.9% | 576.6% | 6.9% | 623.1% |
| Plastic Bag | 0.4 ± 0.1% | 3.7 ± 0.6% | 3.8% | 0.4% | 4.3% |
| Vegetation (0% mc[a]) | 1.2 ± 0.4% | 0.5 ± 0.1% | 0.5% | 1.2% | 1.7% |
| Vegetation (20% mc[a]) | 1.2 ± 0.2% | 0.7 ± 0.3% | 0.7% | 1.2% | 1.9% |
| Vegetation (50% mc*) | 1.0 ± 0.2% | 12.7 ± 0.1% | 14.5% | 1.1% | 15.7% |
| Food | 2.5 ± 0.6% | 13.6 ± 2.8% | 15.7% | 2.5% | 18.7% |
| Combined | 1.1 ± 0.5% | 1.5 ± 0.5% | 1.5% | 1.2% | 2.7% |

[a]mc: fuel moisture content

## 3.5 Discussion: Emission Factors for Solid Waste Open Burning Emission Inventories

One application of EFs is to estimate emission rates for relevant regions in emission inventories (U.S. Epa, 1992). These inventories are used to conduct air quality modeling, track long-term trends, evaluate control strategy effectiveness, and provide offsets for other emitters. For example, emissions avoided by trucking the normally open burned household solid waste to landfill by Sasol's WCI can be estimated as:

$$E_p = AR \times EF_p = \sum_{i=1}^{n} AR_i \times EF_{p,i} \qquad (3)$$

where $E_p$ is total avoided emission of pollutant $p$ (in metric tons per year); $AR$ is the activity rate, i.e., the amount of burned waste avoided in a year (in tons per year); and $EF_p$ is the emission factor (in grams of emissions per gram of waste) of pollutant $p$ from the waste that would otherwise be burned. The subscript $i$ corresponds to values for each waste material $i$ (e.g., paper, textile, plastics, and vegetation). $EF_p$ corresponds to the measured EFs from the combined waste materials; it can also be estimated by summing $EF_{p,i}$ for individual waste materials, weighted by their mass fractions (Fig. 1). $EF_{p,i}$ can be determined from laboratory testing under controlled conditions, and the heterogeneity of waste materials can be accounted for by examining the waste refuse. The separation of flaming and smoldering EFs offers additional flexibility in accounting for burning condition changes. However, it should be cautioned that the burning behaviors differ between separated and combined waste materials, causing emissions to change. Table S5 compares the measured EFs for the combined materials and the values



calculated from $EF_{p,i}$. The calculated EFs agree with the measured values within 10% for $CO_2$ and $NO_x$; however, the calculated EFs for CO and PM are over 50% and 600% higher, respectively. It is possible that more efficient combustion in the combined materials lowered CO and PM emissions as compared to less efficient individual burns, particularly for materials that only smoldered and had high EFs for CO and PM. Additionally, laboratory measured $EF_{p,i}$ or $EF_p$ might differ from field values given the complex waste mixtures and burning conditions. Adjustments to laboratory $EF_{p,i}$ might be needed when estimating real-world $EF_p$. Future studies comparing in situ measurement from a variety of representative real-world burns with laboratory data would assist in establishing adjustment factors.

**4 Conclusions**

This study measured criteria pollutant emissions from simulated combustion of different household solid waste materials representative of those in open burnings in South Africa. EFs vary with waste composition and combustion conditions. Data from this study fill EF gaps for paper, leather/rubber, textile, and food discards burning that have been scarcely reported in the literature. EFs for vegetation and mixed waste materials from this study are within the ranges reported in the literature. These EFs can be used to improve emission inventories for household and municipal solid waste open burning emissions in South Africa and other countries.

Emissions are closely related fuel elemental compositions. Among the tested materials, plastic bags have the highest carbon content and the highest combustion efficiency, leading to the highest EFs for $CO_2$. Textiles have the highest abundances of nitrogen and sulfur, resulting in the highest EFs for $NO_x$ and $SO_2$. Combustion behaviors and emissions are also affected by fuel moisture content. EFs for vegetation with three moisture content: dry (0%), natural (20%), and damp (50%) were measured. Emissions were similar for 0% and 20% moisture content; however, EFs for CO and PM from the vegetation with 50% moisture content are 3 and 30 times, respectively, of those from 0% and 20% moisture content.

This study reports three sets of $EF_s$ (i.e., flaming, smoldering, and entire combustion), which can be applied to estimate emissions based on waste burning characteristics. It also reports EFs for individual and combined waste categories. These data offer flexibility in calculating emission rates depending on waste composition and burning characteristics. However, caution should be exerted when using mass weighted sum of individual waste category EFs to calculate combined waste EFs as the combustion behavior might be different between individual and combined waste materials. This study shows that neglecting the carbon in ash and/or PM may lead to significant overestimation of EFs.

EF data from this study were obtained from controlled laboratory tests simulating real-world open burning conditions. Real-world open burning emissions vary with waste material composition, pile size, packing structure, moisture content, ambient temperature, and wind speed. Such variations are reflected in the wide range of EFs reported in the literature. Although this and past studies agree within reported extremes, laboratory tests are an approximation of real-world variations. The EFs derived from laboratory experiments represent the values obtained under the specific conditions in laboratory tests; adjustment might be needed when real-world burning conditions are very different from laboratory test conditions.



**Data availability.** Data is available upon request.

**Author contributions.** XW, JCC, and JGW designed the study; HF conducted the combustion experiments; XW and HF performed the data analyses and prepared the original paper draft; WC and SDV provided waste materials and resources; all authors reviewed and edited the paper.

**Competing interests.** None.

**Financial support.** This research was partially funded by SASOL and partially by the Desert Research Institute internal funding.

**Acknowledgements:** The authors thank Matthew Claassen of DRI for collecting vegetations for testing.

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
