# Peer review of "Characterization of Gas and Particle Emissions from Open Burning of Household Solid Waste from South Africa"

_EGUsphere, 2023_

## Referee Comment (RC1)

**General comments:**

This manuscript conducts comprehensive laboratory-based experiments to investigate gas and particulate pollutant emissions from open burning of household solid waste. The authors clearly describe the $CO_2$, $CO$, $NO_x$, $SO_2$, and PM emission factors (EFs) of ten types of solid waste materials, and discuss the possible influence factors (e.g., elemental composition, moisture content etc.). Considering different combustion phases (e.g, flaming and smoldering) in their study is a nice feature of this paper. These detailed EFs enhance the database of carbon source emission, which could apparently reduce uncertainties when compiling the emission inventory of carbon for residential combustion sector. This is an extremely important area of research for global carbon budget, and absolutely relevant for the scientific community and decision-makers. I recommend this manuscript can be accepted for publication after addressing the following issues.

**Specific comments:**

**Comment (1) Section 2.3:** Actually, the concentration of pollutant (e.g., $CO_2$, $CO$, PM etc) is always changing during the combustion phase. Here, to obtain the EFs for flaming, smoldering, and entire burning process, $C$ should be the average concentration of pollutant in different burning process. If so, please clarify.

**Comment (2) Section 3.1:** (a). Line 190–191, Figure 3c shows that there are concentration peaks for NO and $NO_2$. It seems that $NO_x$ (= NO + $NO_2$) had the similar peak during 0–400s compared to other pollutants. I think the real-time $NO_x$ concentrations were close to background levels, that were mainly affected by the amount of fuel burned. The low combustion temperatures and low nitrogen content of the fuel should be the major cause of low $NO_x$ EFs. (b). Line 198–199, why authors claim that the lower CO EFs produced by plastic bags was associated with high C and H content? In fact, high C content may lead to high $CO_2$ and CO emission, while the higher MCE would cause the large ratio of $CO_2$ to CO emission.

**Comment (3) Section 3.2:**
(a). Line 211–212, what are the differences between linear regressions with/without intercept in Figure 5? Could you clarify what "other combustion emissions" refers to? I suggest to include all the sample sets for combustion experiments in Figure 5. (b). The color of the filter membrane is interesting, the representative photograph of filter membrane for each waste material could be combined and added in Figure 6 to indicate OC and EC content.

**Comment (4) Section 3.3:** (a). Line 244, Do the authors mean that materials that have both flaming and smoldering phases have similar/comparable EFs? If yes, please present the result of T-test to prove that there is no significant difference between these data. (b). Despite Tables 2 and 3 have all the EFs data from this study and previous literatures, they still need to be described and cited in the main text.

**Comment (5) Section 3.4:** As shown in Table 4, most EF changes without $C_{PM}$ and $CMF_{ash}$ were similar to their content. However, for rubber and plastic bottle burning,

the EF changes without $C_{PM}$ were much higher than $C_{PM}$ content (87.0% vs. 46.5% and 576.6% vs. 85.2%). Could the authors add some explanations that why the larger $C_{PM}$ content caused such greater EF changes.

**Technical Comments:**

**Comment (6):** The current title is bit ambiguous, leading readers to expect the study on global household solid waste combustion. I suggest to explicit that the analysis focuses on the waste materials in South Africa.

**Comment (7) Line 26–27:** Delete "household and" to make the abbreviation (MSW) more clearly.

**Comment (8):** For introduction section, (a) the second and third paragraphs for description of solid waste open burning can be merged; (2) the fourth paragraph related to risk of smoke can be deleted, since there is no discussion of toxicity in this study. I suggest the author could point out the possible link between solid waste open burning emission and global (or South Africa's) carbon budget.

**Comment (9) Line 88:** Delete "organics".

**Comment (10) Line 97–98:** Do "the other categories" refer to glass, metals, and ceramics? If yes, the combined materials seems to be the mixtures of all (not only the other) waste material categories based on their burned mass fractions. Please confirm.

**Comment (11) Line 114:** Replace "Ipcc" with "IPCC".

**Comment (12) Line 117–119:** This paragraph on nitrogen and sulfur contents could be combined with the previous paragraph (both of them are elemental compositions).

**Comment (13) Line 121:** C% content, carbon content, or C content, it is better to write in a uniform way.

**Comment (14) Line 129–132:** How ignited the non-flammable materials? Do the author mean only smoldering emissions were measured until all pollutant concentrations returned to baselines, what about flaming emissions?

**Comment (15) Line 141–142:** Delete the sentence "CO and $CO_2$ concentrations were used to calculate the modified combustion efficiency (MCE) and fuel-based EFs."

**Comment (16):** The instruments (e.g., ELPI, PASS-3) that are not used in this study should not be in Figure 2.

**Comment (17):** Correct the subscripts of $CO_2$ in the formula (1).

**Comment (18) Line 141–142 and:** Replace "~0.9" with "0.9".

**Comment (19) Line 203:** Replace "rubber" with "leather/rubber".

**Comment (20) Line 206:** Replace "~0.92, 0.9, and 0.8" with the MCE values reported in Table 2.

**Comment (21) Line 324:** The descriptions in brackets can be deleted.

---

## Author Comment (AC1)

**Responses to Reviewer 1's Comments**

We thank Reviewer 1's thorough and constructive comments. Please see our response in blue font below.

**General Comments**

1.  *This manuscript conducts comprehensive laboratory-based experiments to investigate gas and particulate pollutant emissions from open burning of household solid waste. The authors clearly describe the $CO_2$, CO, $NO_x$, $SO_2$, and PM emission factors (EFs) of ten types of solid waste materials, and discuss the possible influence factors (e.g., elemental composition, moisture content etc.). Considering different combustion phases (e.g, flaming and smoldering) in their study is a nice feature of this paper. These detailed EFs enhance the database of carbon source emission, which could apparently reduce uncertainties when compiling the emission inventory of carbon for residential combustion sector. This is an extremely important area of research for global carbon budget, and absolutely relevant for the scientific community and decision-makers. I recommend this manuscript can be accepted for publication after addressing the following issues.*
    **Response**: We appreciate the reviewer's positive assessment of the manuscript. The reviewer correctly points out the importance of municipal solid waste (MSW) burning emissions on the carbon emission inventory and global carbon budget. Data from this paper can be used for both air quality management and climate effect assessment. A follow-up paper will further discuss the climate effects of MSW emissions with additional information on black carbon, brown carbon, and aerosol light scattering and absorption properties.

**Specific Comments**

2.  *Comment (1) Section 2.3: Actually, the concentration of pollutant (e.g., CO2, CO, PM etc) is always changing during the combustion phase. Here, to obtain the EFs for flaming, smoldering, and entire burning process, C should be the average concentration of pollutant in different burning process. If so, please clarify.*
    **Response**: Yes, the mean pollutant concentrations were used for EF calculation. The explanation of variables for Eq. (2) is modified as follows:

    > "$C_p$ is the mean plume concentration of pollutant $p$ in g $m^{-3}$ averaged over the calculation period (i.e., flaming, smoldering, or entire combustion process); and $C_{CO}$ and $C_{CO_2}$ are the mean concentrations of $CO_2$ and CO in g $m^{-3}$, respectively. $C_{PM}$ is the mean total caron (TC = OC + EC) concentration in $PM_{10}$ in g $m^{-3}$."

3.  *Comment (2) Section 3.1: (a). Line 190–191, Figure 3c shows that there are concentration peaks for NO and NO2. It seems that NOx (= NO + NO2) had the similar peak during 0–400s compared to other pollutants. I think the real-time NOx concentrations were close to background levels, that were mainly affected by the amount of fuel burned. The low combustion temperatures and low nitrogen content of the fuel should be the major cause of low NOx EFs. (b). Line 198–199, why authors claim that the lower CO EFs produced by plastic bags was associated with high C and H content? In fact, high C content may lead to*

*high CO2 and CO emission, while the higher MCE would cause the large ratio of CO2 to CO emission.*

**Response**:

(a) The $NO_x$ emission description is modified as follows:

> "$NO_x$ concentrations were only slightly above the background levels during the peak emission period, likely due to the low combustion temperatures, low nitrogen content of the plastic bottle (Table S2), and a small amount of material burned."

(b) The reasons for fuels with high C and H content to have lower CO emission is because hydrocarbon fuels have higher combustion efficiencies than fuels with higher oxygen content (e.g., plastic bottles). The sentence is clarified as below:

> "Plastic bags produced the highest $CO_2$ and the lowest CO EFs among all test materials, consistent with the high MCEs due to their high C and H content (Table S2)."

4. *Comment (3) Section 3.2: (a). Line 211–212, what are the differences between linear regressions with/without intercept in Figure 5? Could you clarify what "other combustion emissions" refers to? I suggest to include all the sample sets for combustion experiments in Figure 5. (b). The color of the filter membrane is interesting, the representative photograph of filter membrane for each waste material could be combined and added in Figure 6 to indicate OC and EC content.*

**Response**:

(a) The two different regressions serve different purposes. The regression forced through zero provides a simpler estimate of the $PM_{2.5}/PM_{10}$ ratio. It agrees with the expectation that when $PM_{10}$ is zero, $PM_{2.5}$ should be zero; but it does not agree with the fact that when $PM_{2.5}$ is zero, $PM_{10}$ is not necessary zero. The regression including an intercept reflects experimental uncertainties. Figure 5 does include all sample sets from this study. The phrase "other combustion emissions" refers to those reported in the literature, not from other burns in this study. This sentence is clarified as below:

> "The linear regression slopes indicate that $PM_{2.5}$ constituted ~93% $PM_{10}$, consistent with findings for combustion emissions reported in the literature (e.g., U.S. EPA, 1992; Lemieux, 1997)."

(b) Filter photos are added to Fig. 6 thanks to the reviewer's suggestion.

[Figure]

Revised Fig. 6

5.  *Comment (4) Section 3.3: (a). Line 244, Do the authors mean that materials that have both flaming and smoldering phases have similar/comparable EFs? If yes, please present the result of T-test to prove that there is no significant difference between these data. (b). Despite Tables 2 and 3 have all the EFs data from this study and previous literatures, they still need to be described and cited in the main text.*

    **Response**: (a) An ANOVA test, which is more suitable for multiple group data sets than T-test, was run to test the similarity of $CO_2$ and CO EFs for paper, textile, dry and natural vegetation, and combined waste. The $CO_2$ EFs are similar with a p-value of 0.20, while the CO EFs are statistically different with a p-value <0.05. The description is modified as follows:

    "Except for plastic bags that have high EFs due to high carbon fuel content, total $CO_2$ and CO EFs are relatively consistent for materials that have both flaming and smoldering phases (i.e., paper, textile, dry and natural vegetation, and combined waste), with an RSD of 3% and an ANOVA test p-value of 0.20, in part due to similar fuel carbon contents as shown in Table S2 (RSD = 6%)."

    (b) The references that were included in Table 3 are added to the main text.

6.  *Comment (5) Section 3.4: As shown in Table 4, most EF changes without CPM and CMFash were similar to their content. However, for rubber and plastic bottle burning, the EF changes without CPM were much higher than CPM content (87.0% vs. 46.5% and 576.6% vs. 85.2%). Could the authors add some explanations that why the larger CPM content caused such greater EF changes.*

    **Response**: The impact of CPM on EF can be evaluated using Eq. (2) and its variations. When $C_{PM}$ is not included, the EF is calculated as:

$$EF_{p,i} = \left(CMF_{fuel} - \frac{m_{ash}}{m_{fuel}} CMF_{ash}\right) \frac{C_p}{C_{CO_2}\left(\frac{M_c}{M_{CO_2}}\right) + C_{CO}\left(\frac{M_c}{M_{CO}}\right)} \times 1000 \qquad (a)$$

The EF change relative to Eq. (2) is:

$$\frac{EF_{Eq.(a)} - EF_{Eq.(2)}}{EF_{Eq.(2)}} = \frac{C_{PM}}{C_{CO_2}\left(\frac{M_c}{M_{CO_2}}\right) + C_{CO}\left(\frac{M_c}{M_{CO}}\right)} = \frac{C_{PM}}{Total\ C - C_{PM}} \qquad (b)$$

where $Total\ C = C_{CO_2}\left(\frac{M_c}{M_{CO_2}}\right) + C_{CO}\left(\frac{M_c}{M_{CO}}\right) + C_{PM}$.

For rubber, $\frac{C_{PM}}{Total\ C} = 46.5\%$ (Table 4); plugging this value in Eq. (b) yields the EF change with ash but without $C_{PM}$ to be 87%.

***Technical Comments:***

7. *Comment (6): The current title is bit ambiguous, leading readers to expect the study on global household solid waste combustion. I suggest to explicit that the analysis focuses on the waste materials in South Africa.*
   **Response**: The title is changed to "Characterization of Gas and Particle Emissions from Open Burning of Household Solid Waste from South Africa".

8. *Comment (7) Line 26–27: Delete "household and" to make the abbreviation (MSW) more clearly.*
   **Response**: Revised as suggested.

9. *Comment (8): For introduction section, (a) the second and third paragraphs for description of solid waste open burning can be merged; (2) the fourth paragraph related to risk of smoke can be deleted, since there is no discussion of toxicity in this study. I suggest the author could point out the possible link between solid waste open burning emission and global (or South Africa's) carbon budget.*
   **Response**:
   a) The second and third paragraphs are combined.
   b) The paragraph about health risk is removed.
   c) The impact of waste burning emissions on carbon budget was briefly mentioned in the original text:

   "In addition, open burning emits large amounts of carbon dioxide ($CO_2$) and light absorbing carbon (including black carbon [BC]), two of the largest climate forcers to global warming (Bond et al., 2013; IPCC, 2013)"

   Additional climate link was added:
   "Despite the global health crisis and potential climate impacts caused by uncontrolled solid waste open burning, the quantity of pollutant emissions is uncertain."

10. *Comment (9) Line 88: Delete "organics".*

**Response**: Revised as suggested.

11. *Comment (10) Line 97–98: Do "the other categories" refer to glass, metals, and ceramics? If yes, the combined materials seems to be the mixtures of all (not only the other) waste material categories based on their burned mass fractions. Please confirm.*
**Response**: The reviewer is correct that "the other categories" actually refers to all material categories. This is revised as:

"The combined materials were mixtures of all  categories based on their mass fractions in Fig. 1."

12. *Comment (11) Line 114: Replace "Ipcc" with "IPCC".*
**Response**: The incorrect capitalization for some references was caused by an incorrect setting in the reference management software EndNote. This has been corrected.

13. *Comment (12) Line 117–119: This paragraph on nitrogen and sulfur contents could be combined with the previous paragraph (both of them are elemental compositions).*
**Response**: Revised as suggested.

14. *Comment (13) Line 121: C% content, carbon content, or C content, it is better to write in a uniform way.*
**Response**: Revised as suggested.

15. *Comment (14) Line 129–132: How ignited the non-flammable materials? Do the author mean only smoldering emissions were measured until all pollutant concentrations returned to baselines, what about flaming emissions?*
**Response**: The experiment description is revised as below:

"For nonflammable materials (i.e., leather/rubber, plastic bottle, damp vegetation, and food discards), smoldering emissions were measured when the fuel was heated to 450 °C. Each test started with about 5 minutes sampling of background concentrations and ended when the pollutant concentrations returned to baselines."

16. *Comment (15) Line 141–142: Delete the sentence "CO and CO2 concentrations were used to calculate the modified combustion efficiency (MCE) and fuel-based EFs."*
**Response**: Revised as suggested.

17. *Comment (16): The instruments (e.g., ELPI, PASS-3) that are not used in this study should not be in Figure 2.*
**Response**: We will report ELPI and PASS-3 data in future publications and we think it is a good idea to include them in Fig. 2 for the completeness of the experimental setup and for future references. We added a footnote in Table S4: "Data from ELPI+ and PASS-3 are not included in this paper but will be reported in future publications."

18. *Comment (17): Correct the subscripts of CO2 in the formula (1).*
**Response**: Revised as suggested.

19. *Comment (18) Line 141–142 and: Replace "~0.9" with "0.9".*
    **Response**: Revised as suggested.

---

## Author Comment (AC2)

**Responses to Reviewer 2's Comments**

We appreciate Reviewer 2's positive evaluation of our manuscript and constructive comments. Please see our responses in blue font below.

**General Comments**

1. *Line 74: "However, particle emissions are not often measured in these studies." What studies do the authors refer to here? Yokelson et al. and Jayarathne et al. both use filter- based particle sampling and Stockwell et al. 2016 makes BC measurements of garbage burning. More context is needed here.*
   **Response**: The reviewer is correct that Yokelson et al. (2013), Jayarathne et al. (2018), and Stockwell et al. (2016) measured particle emissions. The sentence "However, particle emissions are not often measured in these studies" is deleted.

2. *Figure 1: Is there a reason for the order of the categories in the bar graph? Just a suggestion for readability, and not a requirement, but it may help if the categories were sorted by mass %.*
   **Response**: There is not a specific reason for the order of the categories in Fig. 1. It happens to be the order of the mass fraction data we received from our South Africa collaborators. We organized the results in the material category order of Fig. 1. While it will improve readability of Fig. 1 to sort categories by mass%, it would require reorganization of the order of all results. Therefore, we prefer to leave the category order in Fig. 1 as is.

3. *Line 85: Can the authors elaborate on if the floor mat is composed of petroleum-based materials (i.e. synthetic) or natural (i.e. cowhide and natural rubber) since these are likely to have different emission factors. The floor mat appears to be synthetic but it should be specified.*
   **Response**: The floor mat appears to be synthetic rubber. This is confirmed by the strong preference of even carbon n-alkanes with the carbon preference index (CPI; the ratio of the sums of odd to even carbon numbers) of particles emitted from the floor mat burning, an indication of petroleum products (Rogge et al., 1993). The floor mat CPI is 0.49, close to that of plastic bottles (0.58) and plastic bags (0.53). On other hand, the CPI from dry vegetation burning emission was 6.01. This data will be published in a future publication. The synthetic specification is added to the floor mat description:

   > "The single synthetic leather/rubber piece (a car floor mat) measured in this study may not be representative of all such materials available elsewhere."

4. *Line 172 Carbon is misspelled 'caron'*
   **Response**: Thanks for catching this. It is corrected.

5. *Line 189 "…likely formed from re-condensation of evaporated plastic molecules". Could the authors elaborate on this suggestion on why the aerosol emissions were highest from plastic bottle burning? Wouldn't condensation of vaporized plastic be true for the plastic bags and synthetic rubber as well? I assume this conclusion comes from the extent of smoldering phase compared to flaming phase?*
   **Response**: Upon heating, plastic materials go through softening, melting, decomposition, and burning stages, depending on the temperature. Bond breakages will likely occur when

plastics are heated to 450 °C, generating smaller molecules. Thermal decomposition of polyethylene and polypropylene (widely used for plastic bags) generates a large amount of volatile and flammable alkanes and alkenes (Bockhorn et al., 1999). These thermal decomposition products are efficiently oxidized to $CO_2$ and CO in the hot flame environment, generating less particle emissions. On the other hand, thermal decomposition of polyethylene terephthalate (PET; widely used for plastic bottles) forms semivolatile carboxylic acid and hydroxyl esters including phthalates as well as non-volatile compounds with interconnected aromatic rings (Holland and Hay, 2002; Sovová et al., 2008). These semivolatile decomposition products quickly cool after leaving the heating zone and recondense into particles. The description of plastic bottle particle emission is revised as below:

> "However, PM emissions were the highest among all the waste materials. The strong plastic odor and light-yellow colored sticky particles were likely formed from condensation of semi-volatile thermal decomposition products, such as carboxylic acids and hydroxyl esters including phthalates (Holland and Hay, 2002; Sovová et al., 2008)."

6. *Figure 4: Can the authors explain why nitrogen oxide emissions are observed from plastic bag combustion and not from plastic bottle burning?*
   **Response**: $NO_x$ can be formed from two main mechanisms in the combustion process relevant to this paper: 1) oxidation of fuel-bound nitrogen (N); and 2) oxidation of combustion air nitrogen ($N_2$) in the high temperature flame (Hill and Douglas Smoot, 2000). It is postulated that the fuel nitrogen is first released mainly as tar molecules and light gases (e.g., hydrogen cyanide [HCN] and ammonia [$NH_3$]) when volatilized upon heating. These molecules then react with oxygen ($O_2$) under high temperatures to form $NO_x$. Because plastic bottles only smolder, the volatized molecules quickly cool after leaving the heating zone and have little chance to react with $O_2$ to form $NO_x$. On the other hand, in the flaming combustion of plastic bags, the combustion temperature is much higher. The volatilized molecules have more time to react with oxygen in the flame to form $NO_x$; some $NO_x$ can also form from the oxidation of air $N_2$ in the high temperature flame. The relative $NO_x$ contributions from these two mechanisms are unknown. The following sentence is added to the text:

> "Due to the higher combustion temperatures, $NO_x$ concentrations during plastic bag burning were also higher than those in plastic bottles burning."

7. *Figure 6: I find it interesting that the OC and EC PM2.5 mass fractions are similar for the plastic bags and combined burning. Does this suggest that when plastic bags are contained in garbage the plastic burning dominates the total emissions? Or does it suggest that when plastic bags are contained in the garbage it increases the higher efficiency flaming emissions of the combined refuse?*
   **Response**: We think plastic bag increased the combustion efficiency of the combined materials. As shown in Table S5 and discussed in Section 3.5, burning behaviors differ between separated and combined waste materials, causing emissions to change. We think the similar high combustion efficiency of plastic bags and combined materials is the main cause for the similar OC and EC abundances in $PM_{2.5}$.

References:

Bockhorn, H., Hornung, A., Hornung, U., Schawaller, D. (1999). "Kinetic study on the thermal degradation of polypropylene and polyethylene." *Journal of Analytical and Applied Pyrolysis* 48 (2):93-109. https://doi.org/10.1016/S0165-2370(98)00131-4

Hill, S.C., Douglas Smoot, L. (2000). "Modeling of nitrogen oxides formation and destruction in combustion systems." *Progress in Energy and Combustion Science* 26 (4):417-458. https://doi.org/10.1016/S0360-1285(00)00011-3

Holland, B.J., Hay, J.N. (2002). "The thermal degradation of PET and analogous polyesters measured by thermal analysis–Fourier transform infrared spectroscopy." *Polymer* 43 (6):1835-1847. https://doi.org/10.1016/S0032-3861(01)00775-3

Rogge, W.F., Hildemann, L.M., Mazurek, M.A., Cass, G.R., Simoneit, B.R.T. (1993). "Sources of fine organic aerosol. 2. Noncatalyst and catalyst-equipped automobiles and heavy-duty diesel trucks." *Environmental Science & Technology* 27 (4):636-651.

Sovová, K., Ferus, M., Matulková, I., Španěl, P., Dryahina, K., Dvořák, O., Civiš, S. (2008). "A study of thermal decomposition and combustion products of disposable polyethylene terephthalate (PET) plastic using high resolution fourier transform infrared spectroscopy, selected ion flow tube mass spectrometry and gas chromatography mass spectrometry." *Molecular Physics* 106 (9-10):1205-1214. 10.1080/00268970802077876